# Localized skin inflammation during cutaneous leishmaniasis drives a chronic, systemic IFN-γ signature

**Camila Farias Amorim**[1], **Fernanda O. Novais**[1¤], **Ba T. Nguyen**[1], **Mauricio T. Nascimento**[2,3], **Jamile Lago**[2,3], **Alexsandro S. Lago**[2,3], **Lucas P. Carvalho**[2,3], **Daniel P. Beiting**[1]*, **Phillip Scott**[1]*

**1** Department of Pathobiology, School of Veterinary Medicine, University of Pennsylvania, Philadelphia, United States of America, **2** Serviço de Imunologia, Complexo Hospitalar Universitário Professor Edgard Santos, Universidade Federal da Bahia, Salvador, Brazil, **3** Laboratório de Pesquisas Clínicas do Instituto de Pesquisas Gonçalo Moniz–Fiocruz, Salvador, Brazil

¤ Current address: Department of Microbial Infection and Immunity, College of Medicine, The Ohio State University, Columbus, Ohio, United States of America
* beiting@upenn.edu (DPB); pscott@upenn.edu (PS)

**Data Availability Statement:** Raw sequence data of peripheral blood and lesions from CL patients as well as the clinical metadata are available on the

## Abstract

Cutaneous leishmaniasis is a localized infection controlled by CD4+ T cells that produce IFN-γ within lesions. Phagocytic cells recruited to lesions, such as monocytes, are then exposed to IFN-γ which triggers their ability to kill the intracellular parasites. Consistent with this, transcriptional analysis of patient lesions identified an interferon stimulated gene (ISG) signature. To determine whether localized *L. braziliensis* infection triggers a systemic immune response that may influence the disease, we performed RNA sequencing (RNA-seq) on the blood of *L. braziliensis*-infected patients and healthy controls. Functional enrichment analysis identified an ISG signature as the dominant transcriptional response in the blood of patients. This ISG signature was associated with an increase in monocyte- and macrophage-specific marker genes in the blood and elevated serum levels IFN-γ. A cytotoxicity signature, which is a dominant feature in the lesions, was also observed in the blood and correlated with an increased abundance of cytolytic cells. Thus, two transcriptional signatures present in lesions were found systemically, although with a substantially reduced number of differentially expressed genes (DEGs). Finally, we found that the number of DEGs and ISGs in leishmaniasis was similar to tuberculosis–another localized infection–but significantly less than observed in malaria. In contrast, the cytolytic signature and increased cytolytic cell abundance was not found in tuberculosis or malaria. Our results indicate that systemic signatures can reflect what is occurring in leishmanial lesions. Furthermore, the presence of an ISG signature in blood monocytes and macrophages suggests a mechanism to limit systemic spread of the parasite, as well as enhance parasite control by pre-activating cells prior to lesion entry.

NCBI Gene Expression Omnibus (GEO, accession GSE162760 and GSE127831, respectively).

**Funding:** PS is funded by the National Institutes of Health (NIH) R01-AI-149456 (https://grants.nih.gov/grants/funding/r01.htm). LPC is funded by the National Institute of Allergy and Infectious Diseases AI 136032 (https://www.niaid.nih.gov). The funders had no role in study design, data collection and analysis, decision to publish, or preparation of the manuscript.

**Competing interests:** The authors have declared that no competing interests exist.

## Author summary

Cutaneous leishmaniasis caused by the protozoan *Leishmania braziliensis* exhibits two dominant inflammatory responses in cutaneous lesions: Interferon-γ (IFN-γ)-mediated signaling, which promotes parasite control, and cytolysis mediated by cytotoxic CD8+ T and NK cells, which promotes increased pathology. To determine if these responses were limited to cutaneous lesions, we performed RNA-seq on the blood of cutaneous leishmaniasis (CL) patients, and detected both transcriptional signatures in the peripheral blood. The presence of interferon stimulated genes, as well as circulating IFN-γ, suggests that protective immune responses are not limited to the lesion site, but are occurring systemically. This may be one mechanism to ensure optimal control of the parasites, both by limiting their systemic spread and by pre-activating cells to kill the parasites prior to entry into the lesions. The cytolytic transcriptional signature was uniquely detectable in the blood of *L. braziliensis* patients when compared to the blood of patients with tuberculosis (TB) or malaria, further emphasizing the importance of this pathway in cutaneous leishmaniasis. Taken together, these data suggest that this localized infection has a systemic component that may have an impact the development of the disease.

## Introduction

Cutaneous leishmaniasis is a localized skin infection caused by the protozoan parasite *Leishmania* spp. and transmitted to the host through the bite of infected sand flies. There are no vaccines for cutaneous leishmaniasis and the standard treatment with pentavalent antimony has toxic side effects and is associated with high rate of failure in endemic areas [1–5]. While the clinical forms of the disease are quite diverse, control of the parasite largely depends upon the production of IFN-γ by CD4+ Th1 cells [6,7]. T cells within lymph nodes that drain the site of infection proliferate and differentiate into Th1 cells due to the presence of IL-12. These T cells then transit to the local lesion site, where they produce IFN-γ, leading to macrophage activation and parasite control. During the initial stages of infection recruited monocytes provide a safe haven for parasites [8], but as the infection progresses and immunity develops, infiltrating monocytes are activated by IFN-γ produced by both effector T cells and skin resident memory T cells and contribute to protection [9,10].

Untargeted transcriptomic approaches have expanded our understanding of many infectious diseases. For example, a transcriptional signature in the peripheral blood distinguishes patients undergoing a reversal reaction in leprosy [11], and genes associated with disease progression in tuberculosis (TB) have been identified in a series of studies [12–16]. Several studies also identified the transcriptional response associated with pathological cerebral malaria in humans and experimental mouse models [17]. Recently, peripheral blood gene signature was found to be predictive for treatment outcome in Ethiopian patients with visceral leishmaniasis and HIV [18]. Transcriptional analysis of lesions from patients infected with *L. braziliensis* found a strong ISG signature, as well as a cytotoxic signature [19–21]. These studies in patients, and parallel studies in murine models, defined an immunopathological pathway initiated by cytolysis, leading to NLRP3 inflammasome activation and release of pro-inflammatory IL-1β [22–24]. Importantly, blockade of either NLRP3 or IL-1β ameliorated pathology in murine models suggesting that host-directed therapies may be useful in lessening disease in patients. The significance of this pathway in patients was confirmed by correlating cytolytic gene expression with treatment outcome [19,20,22,24]. An analysis of transcriptional responses in the blood of patients with visceral leishmaniasis identified expression of both an ISG and

cytotoxic signature [25], but no studies have fully examined the transcriptional profile in the blood of patients with cutaneous leishmaniasis (CL) [26], where disease is highly localized to the skin lesion.

To investigate the systemic transcriptional signatures in cutaneous leishmaniasis and understand what insights these signatures can provide in understanding the immunological responses developed in this disease, we performed RNA sequencing (RNA-seq) in the peripheral blood of *L. braziliensis*-infected patients. We observed a consistent and significant monocyte and macrophage-associated Interferon-stimulated gene (ISG) signature in the blood of CL patients. Additionally, a cytotoxic signature including the expression of *GZMB*, *PRF1*, and *GNLY* was also evident. To understand these responses relative to other infections we compared our results with published data from human tuberculosis and malaria. Although a ISG response was common to all three infections, the cytolytic signature was unique to leishmania. Thus, our results indicate that although cutaneous leishmaniasis is a local skin infection, a systemic transcriptional profile is present in patients and mirrors what is seen in the lesion. Furthermore, our studies suggest that a systemic immune response can activate cells prior to entry into leishmania lesions, thus potentially enhancing parasite control.

## Methods

### Ethics statement

This study was conducted according to the principles specified in the Declaration of Helsinki and under local ethical guidelines (Ethical Committee of the Medical School, UFBA, Salvador, Bahia, Brazil, and the University of Pennsylvania Institutional Review Board). Informed consent for the collection of samples and subsequent analysis was obtained.

### Study design

Fifty patients with cutaneous leishmaniasis (CL patients), with a characteristic skin lesion and positive PCR for *L. braziliensis* participated in this study. Exclusion criteria included previous anti-leishmanial treatment, individuals under 18 years old, pregnancy, or the presence of other comorbidities. Before treatment, 3 ml of peripheral blood was collected from CL patients and 14 endemic non-*L. braziliensis* infected controls (healthy subjects, HS) in Corte de Pedra, Bahia, Brazil, and were stored in Tempus Tubes (Applied Biosystems) at -20°C. Eighteen patients from the 50 patients who had peripheral blood drawn also had a 4mm biopsy collected from the border of their cutaneous lesions. 7 biopsy samples were also collected from the skin non-infected individuals to serve as controls. Patients were given standard of care treatment (daily intravenous injections of pentavalent antimony; 20mg/kg/day for 20 days). At day 90, after the start of treatment, patients were evaluated for lesion resolution. Cure was defined as re-epithelialization of lesions in the absence of raised borders. Patients with active lesions at 90 days were defined as failing treatment and were given an additional round of chemotherapy.

### Measurement of IFN-γ in the serum of CL patients

Serum from 17 randomly selected CL patients and 10 HS was obtained from peripheral blood by centrifugation and stored at -80°C. Concentration of IFN-γ was determined by Enzyme-linked immunosorbent assay (ELISA—BD biosciences), according to the manufacturer's instructions. The results are expressed in pg/ml.

### Processing peripheral blood of CL patients for RNA-seq

Blood samples were collected and stored in Tempus tubes at -80C and shipped to the University of Pennsylvania, where RNA extraction, cDNA library preparation, and RNA sequencing were performed. RNA was extracted using the Tempus Spin RNA Isolation Reagent Kit (Applied Biosystems), and RNA quality and quantification were assessed using a Tapestation 4200 (Agilent) and Qubit 3 (Invitrogen), respectively. Whole-transcriptome sequencing libraries were prepared with the TruSeq Stranded Total RNA with Ribo-Zero Globin (Illumina) and sequenced on an Illumina NextSeq 500 to produce 75 bp paired-end reads with a mean sequencing depth of 32 million reads per sample. Raw reads were mapped to the human cDNA reference transcriptome (Ensembl, GRCh38 release 97) using Kallisto, version 0.46.0 [27].

### RNA-seq data analysis and visualization

All analyses and visualizations were carried out using the statistical computing environment R version 3, RStudio version 1.2.5042, and Bioconductor version 3.11 [28]. Transcript-level counts were summarized to genes using the TxImport package [29] and human gene annotations from the biomaRt package [30]. Data from CL patients was generated as described above. Publicly-available RNA-seq data from tuberculosis (21 patients with active diseases and 12 healthy subjects from the London cohort) and malaria (65 adult patients with uncomplicated malaria, and 16 healthy subjects) were downloaded from the Sequence Read Archive (SRA). Data were normalized using the Trimmed Mean of M-values (TMM) method in EdgeR package [31]. Genes with < 1 count per million (CPM) in *n* of the samples, where n is the size of the smallest group of replicates, were filtered out. Normalized, filtered data were variance-stabilized using the VOOM function in Limma [32], and batch effect correction was carried out using the Combat function from the sva package [33] or intercepting for the batch factor in the model matrix for differential gene expression (DGE) [32]. DGE testing with Benjamini-Hochberg correction for multiple testing was carried out using Limma. [32]. Permutational multivariate analysis of variance (PERMANOVA) was calculated using the vegan package [34]. GO analysis was carried out using the gprofiler2 package [35]. Gene Set Enrichment Analysis (GSEA) was carried out using GSEA software (Broad Institute, version 4.0.2) and Reactome, KEGG and Biocarta pathways databases [36]. Single sample GSEA (ssGSEA) was carried out using the GSVA package [37] and the 51-gene peripheral blood leishmanial signature. The immunedeconv package [38] was used to estimate cell abundances from RNAseq data using the MCP-counter method [39]. Type I- or II IFN-stimulated genes were identified using InterferomeDB v2.0 [40]. Spearman's rho (ρ) correlations were performed in the R statistical environment using the ggpubr and Hmisc package. Wilcoxon rank-sum test was performed using the ggpubr and stats package. Coefficient of variation (cV) was calculated as described previously [19].

## Results

### Transcriptional profiling identifies an Interferon-stimulated gene (ISG) signature in the peripheral blood of patients infected with *Leishmania braziliensis*

To investigate the systemic transcriptional signatures associated with CL, we performed RNA-seq on the peripheral blood collected from 50 CL patients and 14 healthy subjects (HS). Peripheral blood gene expression distinguished CL patients from HS by principal component analysis (PCA; PERMANOVA, F statistic = 0.006) (Fig 1A). GSEA using the Reactome

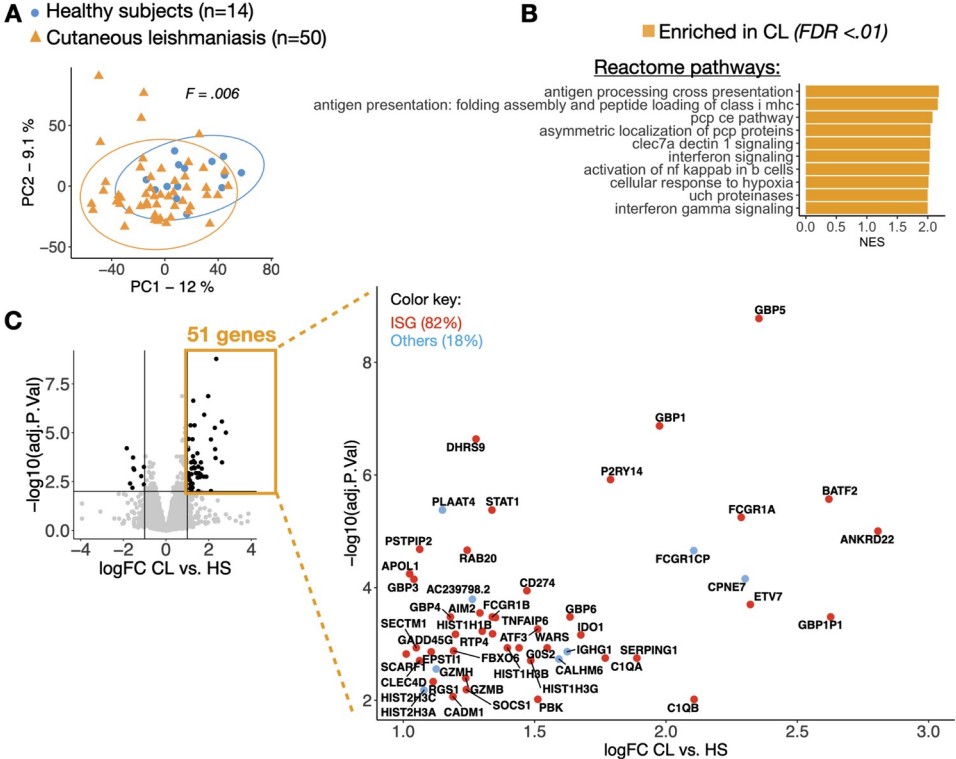

**Fig 1. Transcriptional profiling identifies an Interferon-stimulated gene (ISG) signature in the peripheral blood of patients infected with *Leishmania braziliensis*.** (A) Principal component analysis showing principal component 1 (PC1) and PC2 for RNA-seq data from the peripheral blood of CL patients (yellow triangles, n = 51) and HS (blue circles, n = 14). A PERMANOVA statistical test was used to calculate distances between the groups, Pr(>F) = .006. (B) Gene set enrichment analysis (GSEA) showing normalized enrichment scores (NES) of the top 10 pathways from the Reactome Pathway Database enriched in the peripheral blood of CL patients relative to HS. (C) Volcano plot highlighting overrepresented genes (n = 51) in the peripheral blood from CL patients relative to healthy subjects, FDR≤0.01 and logFC≥1. The Interferome database was used to identify Interferon-stimulated genes (ISGs), represented in red circles (n = 42, 82%). Other genes are colored in blue (n = 9, 18%).

Pathway Database revealed the Interferon-gamma signaling within the top 10 pathways enriched in CL patients (Fig 1B). Other pathways significantly enriched in CL relative to HS were also associated with Interferon signaling, although annotated as the intracellular antigen processing: ubiquitination and proteasome degradation; antigen presentation: folding, assembly and peptide loading of class I MHC; C-type lectin receptors (clec7a dectin 1) signaling; activation of NF-κB in B cells, cellular responses to external stimuli (hypoxia) and the metabolism of proteins (UCH proteinases), FDR≤0.01 (Fig 1B). No signatures were significantly enriched in the blood of HS relative to CL, FDR ≥0.01. To further investigate specific-gene signatures present in the peripheral blood of CL patients, we carried out a differential gene expression (DGE) analysis between CL relative to HS. We observed that 60 genes were differentially expressed between the two groups. Included in the 51 genes upregulated in CL compared to HS (FC≥2, and FDR≤.01) (Fig 1C), were interferon-responsive genes, such as guanylate-binding proteins (*GBP1-GBP6* and *GBP1P1*), *STAT1*, *SOCS1*, *WARS*, Fc gamma receptor genes (*FCGR1A*, *FCGR1B*, *FCGR1CP*), the transcription factor *ATF3*, and *GZMB* that encodes the cytolytic granule granzyme B. Gene Ontology (GO) enrichment analysis of these 51 genes confirmed a significant enrichment for cellular responses to type II interferon (FDR≤0.0001.)

### An Interferon-stimulated type II gene (ISG) systemic signature mirrors the local immune response in the CL lesion

Given the fact that type I interferon (IFN-a/b) and type II interferon (IFN-γ) signal through similar downstream transcriptional responses, and given multiple reports of both interferon responses occurring during protozoan parasite infections [42,43], we next used the Interferome database [40] to more precisely annotate these 51 genes with respect to interferon signaling. 42 out of the 51 genes (82%) were confirmed as Interferon-stimulated genes (ISGs) in Interferome. From this set of 42 ISGs, 5 were annotated as exclusively type II-ISGs (*C1QB*, *PKB*, *DHRS9*, *GADD45G*, and *CADM1*), none were exclusively type I-ISGs, and the remaining 37 respond to both type I and type II interferons (Fig 2A and 2B). These data suggest that the ISG signature in the blood of CL patients may be driven by IFN-γ.

Additionally, to understand the degree to which gene expression changes in the blood of CL patients reflect what is happening at the site of infection, in the leishmanial skin lesion, RNA-seq was performed on lesion biopsies obtained from a subset of the same patients analyzed above. As expected, the number of differentially expressed genes in lesions was much higher than in the blood (4269 vs. 60 total DEG, respectively; FDR≤0.01, and logFC≥1) (Fig 2A). Similarly, 1774 and 42 ISGs were upregulated in lesion and blood, respectively (Fig 2A and 2B). Arbitrary thresholds imposed during DEG analysis can result in poor performance when comparisons are made across tissues where the magnitude of the response differs. In contrast, the threshold-free approach employed by GSEA showed a similar enrichment for IFN signatures in both blood and lesions from CL patients (S1 Fig). Since the number of exclusive type-II ISGs was greater than the number of exclusive type-I ISGs in the blood, and considering that there were no exclusive type-I ISGs upregulated in the blood, we hypothesize that transcriptional changes associated with IFN-γ predominant in CL. To determine if exposure to IFN-γ might be occurring in the blood, we measured IFN-γ protein in CL patients and found elevated levels in the serum (Fig 2C). We also observed a significant correlation between serum levels of IFN-γ protein and the peripheral blood ISG signature in CL (ρ = .79 and P = .0001) (Fig 2D). Taken together, these results show a systemic Th1 response in CL that mirrors the ISG signature observed at the site of the infection in the skin.

### A cytotoxicity transcriptional signature is observed in the peripheral blood of patients infected with L. *braziliensis*

While the gene expression program observed in the peripheral blood of CL patients was enriched for a type II ISG signature, we noticed that *GZMB* and *GZMH* were also significantly upregulated in our DEG analysis (FDR≤0.01 and FC≥2). These genes encode proteins associated with cytolysis by CD8+ cytotoxic T cells (CTL) and NK cells. Cytolytic responses are associated with increased pathology in CL lesions and predict treatment failure [19,22,24]. Therefore, we explored this signature in the peripheral blood of CL patients. Taking a candidate gene approach, we observed that *GZMA*, *PRF1*, and *GNLY* were also upregulated in the blood of CL patients compared to HS (P < .05) (Fig 3A). Additionally, three cytotoxicity-related signatures from the Biocarta and KEGG Pathway Databases (NK cells pathway, NK cell-mediated cytotoxicity pathway and CTL pathway) were enriched in the peripheral blood of CL patients relative to HS (FDR = 0.01, 0.03 and 0.06, respectively) (Fig 3B). Leading edge analysis of the peripheral blood DEGs from CL patients showed that Killer cell lectin-like receptor genes from the CD94/NKG2 family, including KLRC1-3, and KLRD1, and components of the cytolytic and antigen presentation and costimulation machinery (GZMB, PRF1, and HLA-A) were enriched and shared between at least two of the signatures (Fig 3B). These

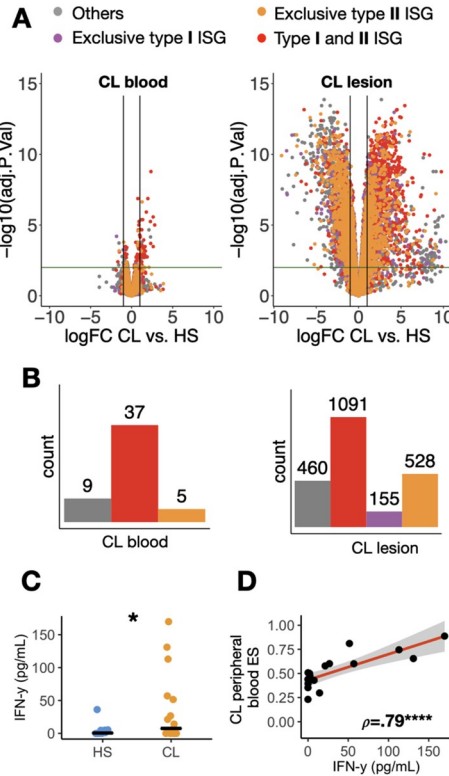

**Fig 2. An Interferon-stimulated type II gene (ISG) systemic signature parallel the local immune response in the CL lesion.** (A) Volcano plots with differentially expressed genes in the peripheral blood (left) and lesion (right) in CL patients relative to HS. Genes were colored according to their ISG classification: gray, not an ISG; purple, an ISG exclusively induced by type I Interferons; red, an ISG exclusively induced by type II Interferon; and yellow, an ISG induced by both type I and II Interferons. Horizontal and vertical lines indicate FDR = 0.01 and logFC = 1, respectively). (B) Histograms show the number of overrepresented type I and type II ISGs from volcano plots above in A. (C) IFN-γ levels in the serum of 17 CL patients infected with *L. braziliensis* and ten healthy subjects (HS). Wilcoxon rank-sum test was used for statistical analysis. *P<0.05. (D) A Spearman correlation between the 51-gene peripheral blood leishmanial signature enrichment score (ES) and the levels of IFN-γ in the serum of CL patients. ρ, Spearman's rho correlation coefficient; ****P>0.0001.

results indicate that, in addition to the strong IFN-γ and ISGs, a cytotoxicity signature is detected in the blood of CL patients.

The expression of cytolytic granule genes in CL lesions is dramatically upregulated compared to healthy skin, and the CD8+ T cell-mediated cytotoxicity has a critical pathogenic role in lesion development [22]. We asked whether the magnitude of cytolytic granule gene expression in the lesion was correlated to the magnitude in the blood in the same individual by analyzing 18 samples from CL patients who had paired lesion and blood RNA-seq data; however, we did not observe a significant correlation (P>.05) (S2 Fig).

## The peripheral blood ISG signature in CL patients is correlated with monocytes

To investigate whether specific cell subtypes in the blood were associated with the ISGs and cytotoxic signature observed systemically in CL patients, we used the MCP-counter method to estimate the abundance of different cell types in our blood RNA dataset [39]. Increased abundance of NK cells, CTLs, monocytes and monocytes/macrophages-like cells were observed in

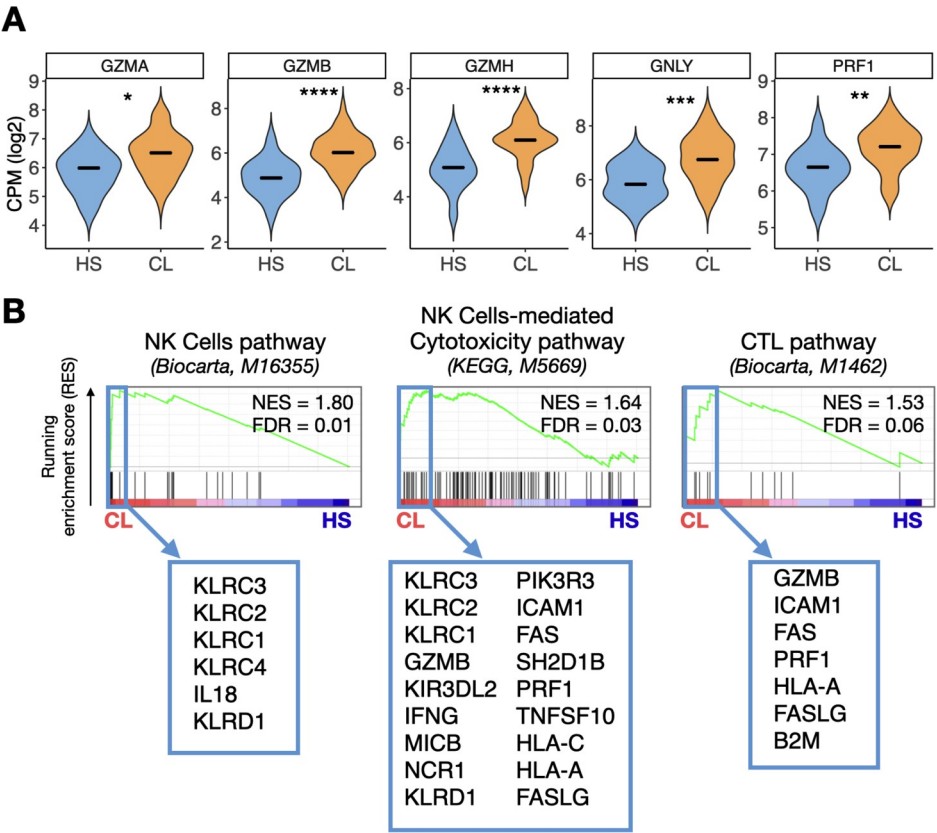

**Fig 3. A cytotoxicity transcriptional signature is observed in the peripheral blood of patients infected with *L. braziliensis*.** (A) Expression of genes encoding cytolytic granules (*GZMA*, *GZMB*, *GZMH*, *GNLY*, and *PRF1*) in the peripheral blood of HS and CL patients. Wilcoxon rank-sum test was used for statistical analysis, *P<0.05, **P<0.01, ***P<0.001 and ****P < .0001. Gene expression is represented as counts per million (CPM) in log2 scale. (B) GSEA enrichment plots showing three cytotoxicity-related pathways from the Biocarta and KEGG Pathway Databases enriched in the CL peripheral blood relative to HS. Pathway name is indicated in parenthesis, next to the pathway database source. Light blue box indicates genes included in the Leading Edge subgroup from each pathway that were enriched and differentially expressed in CL relative to HS. Genes are shown in ranked order according to their running enrichment score. NES, normalized enrichment score; FDR, false discovery rate.

the blood samples from CL patients compared to HS (P<0.05) (Figs 4A and S3). The majority of the ISGs were positively correlated with MCP-counter cell type abundance scores for monocytes and macrophage/monocyte-like cells (Fig 4B), suggesting that these cells are the potentially the ones that are mainly being activated by the circulating IFN-γ. Interestingly, we also observed a significant positive correlation between *GZMB* and *GZMH* expression, and the abundance of NK cells and CTLs (P<0.05) (Fig 4B). Additionally, there was also a significant positive correlation between *GZMA*, *PRF1*, *GNLY* and the abundance of NK cells and CTLs (P<0.05) (S4 Fig), suggesting that the systemic cytolytic signature observed in CL patients is associated with these cytotoxic cells. We previously reported that the expression of cytolytic genes predicted treatment failure in CL patients receiving standard-of-care therapy with pentavalent antimony [19]. Unfortunately, neither systemic expression of *GZMB*, *GNLY*, and *PRF1* expression nor increased abundance of CTLs or NK cells in the blood (S5B and S5C Fig) were predictive of treatment outcome. In addition, PCA analysis of our blood data and enrichment score for our 51-gene peripheral blood signature failed to distinguish patients that cured versus failed antimony treatment (P>0.05) (S5D and S5E Fig). All of these patients were

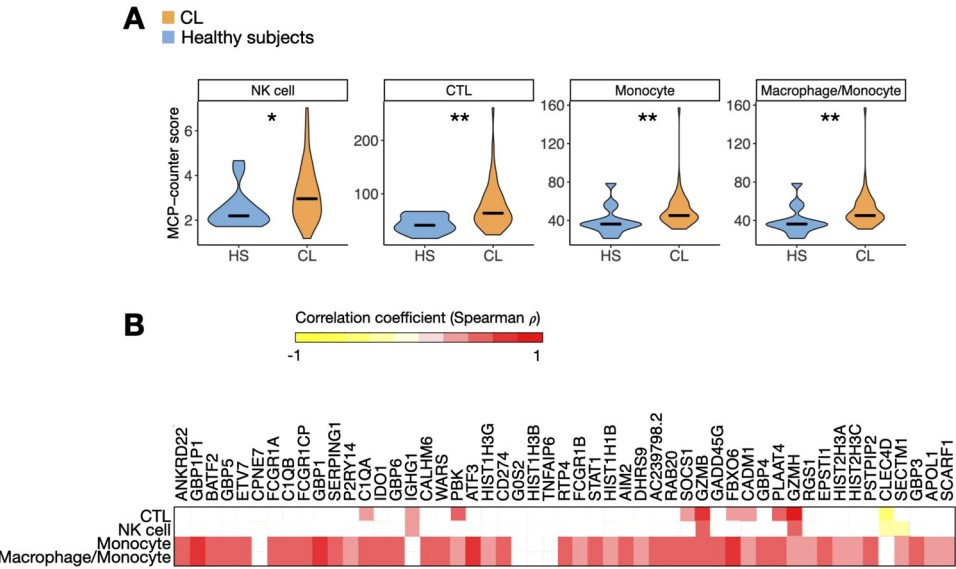

**Fig 4. Peripheral blood cytotoxicity and ISG signatures are associated with CTL/NK cells and monocytes/monocyte-like macrophages, respectively.** (A) Median of MCP-counter abundance scores for cell types that were differentially enriched between CL patients and HS. NK cells, cytotoxic lymphocytes (CTL), monocytes and macrophage/monocytes were differentially enriched by Wilcoxon rank-sum test, P<0.05. (B) A correlation matrix between the MCP-counter abundance scores and the expression of the 51 genes overexpressed in the blood of CL patient samples, in log2 CPM. Spearman's rho (ρ) correlation coefficient ranges from 1 (red) to -1 (yellow). Correlations where P<0.05 are not represented in the plot (blank spots).

treated with antimony after sample collection, and therefore we cannot rule out that differential gene expression in the blood would have influenced disease progression or severity in the absence of treatment.

## Tissue localization may impact the quality and quantity of the peripheral blood response during infection

To understand our results in the context of other infectious diseases, we compared our findings with data of studies of two other intracellular pathogens. Two publicly available RNA-seq datasets of the whole blood from adult patients with active tuberculosis (TB, n = 21) [41] and uncomplicated malaria (n = 65) [14] were analyzed similarly to the CL blood dataset to investigate the transcriptional signatures when compared to controls (n = 12 for TB, and n = 16 for malaria). This also allowed us to compare the blood signature in a disease with a relatively localized infection (TB) and one where the infection is systemic (malaria). Principal component analysis of the TB and malaria datasets showed that infection contributed to the largest source of variation in each dataset (Fig 5A). Previous analyses of these datasets found an upregulation of ISGs [12–14], and here we sought to compare the relative magnitude of the type I and II ISG transcriptional changes between the CL, TB and malaria dataset by performing DGE analysis. While the 42 ISGs were upregulated in the blood of CL patients (Fig 2A and 2B), 133 ISGs were upregulated in the blood of patients with active TB, in which 85% is composed by type I and type II ISG (Fig 5B). These results are consistent with the previous analysis of this dataset [13,14]. In contrast, malaria was marked by a much larger number of DEGs that included 1,288 ISGs, as well as many genes not annotated as part of the Interferome (Fig 5B) (FDR≤0.01 and FC≥2). In addition, ISGs observed in TB and CL were more strongly induced in the blood of malaria patients. These results indicate that *L. braziliensis* and TB, perhaps due to the tissue-specific nature of these infections, elicit quantitatively and qualitatively similar

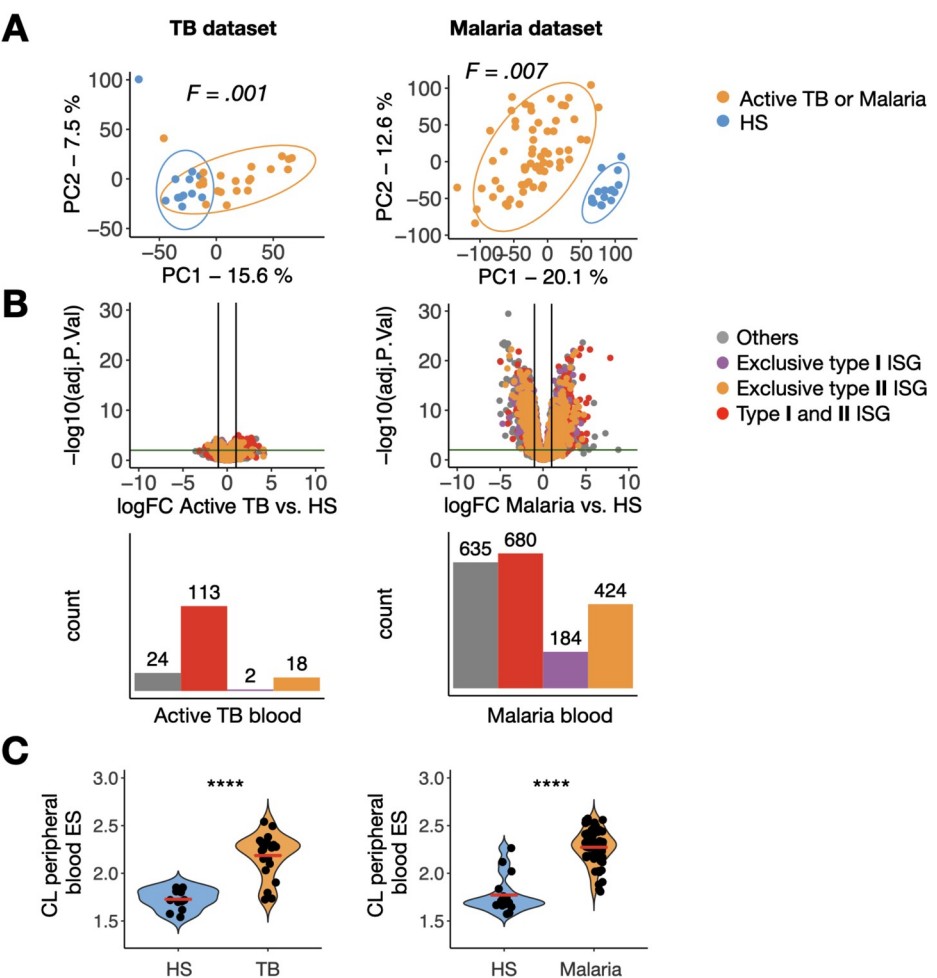

**Fig 5. A peripheral blood signature of cytotoxicity distinguishes CL from malaria and TB patients.** (A) Principal component analysis showing principal component 1 (PC1) and PC2 for RNA-seq data from the peripheral blood of patients with active tuberculosis (left, n = 21) and malaria (right, n = 65) and study controls (HS in blue, n = 12 and n = 16, respectively). (B) Top, Volcano plots showing DEGs in TB and malaria, with type I and II ISGs indicated (horizontal and vertical lines denote FDR = 0.01 and logFC = 1, respectively). Bottom, the number of overrepresented genes in the blood of active TB and malaria patients. (C) CL peripheral blood enrichment score per sample in the TB dataset and malaria dataset by ssGSEA. Wilcoxon rank-sum test was used for statistical analysis. ****P<0.0001; ns, non-significant.

systemic ISG, although there were more ISGs induced in TB. In contrast, in malaria there was a 10-fold increase in ISGs compared to TB and CL. Single sample GSEA (ssGSEA) allowed us to test each TB and malaria RNA-seq sample for enrichment of the 51-gene peripheral blood leishmanial signature. This analysis showed that the ISGs observed in the peripheral blood of CL patients were also induced in the majority of patients with active TB and malaria relative to their own study controls (P<0.001) (Fig 5C), indicating that this ISG signatures is likely conserved amongst Th1 inducing pathogens.

## A peripheral blood signature of cytotoxicity distinguishes CL from malaria and TB patients

Our observation that the peripheral blood ISG response observed in CL patients was also induced to similar or higher levels in TB and malaria, raised the question of whether there

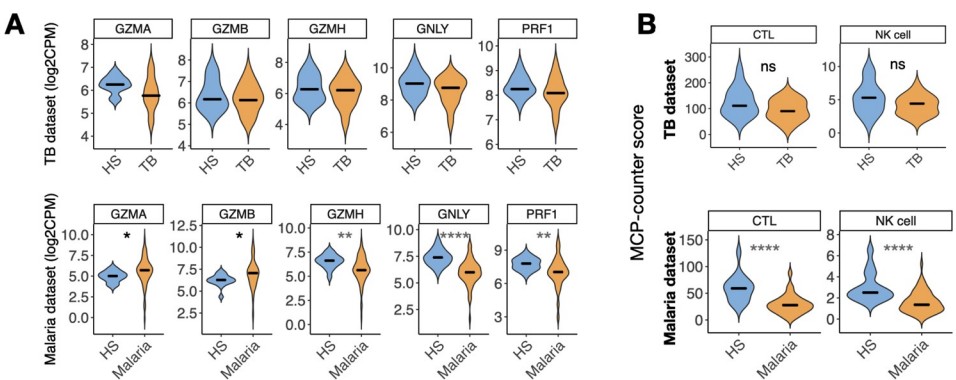

**Fig 6. The CTL/NK cells signature observed in the peripheral blood of CL is unique when compared to the blood of patients with active TB and malaria.** (A) Expression of genes encoding associated with cytolysis (*GZMA*, *GZMB*, *GZMH*, *GNLY*, and *PRF1*) in the peripheral blood of patients with active TB (top) and malaria (bottom) and study controls (healthy subjects, HS). Wilcoxon rank-sum test was used for statistical analysis, *P<0.05, **P<0.01, ****P<0.0001. CPM, counts per million log2 scale. (B) Median of MCP-counter scores between patients with active TB or malaria (yellow) and study controls (blue). Wilcoxon rank-sum test was used for statistical analysis, **P<0.01, ****P<0.0001.

were any aspects of the peripheral blood program that were unique to CL. To address this question, we carried out GSEA on the TB and malaria datasets and tested for enrichment of our CL signature. Although most of the CL genes fell within the leading edge for both the TB and malaria datasets, indicating that they are strongly induced during all three infections, several were either not part of the leading edge or were actually downregulated during either TB or malaria. Amongst these were two cytolytic genes (*GZMH* and *GZMB)* (S6 Fig). To investigate this further, we expanded our analysis to include other components of the cytolytic machinery and found that *GZMA*, *GNLY*, and *PRF1* were not upregulated in TB compared to HS (P>0.05) (Fig 6A). Similarly, in malaria these genes were either only modestly upregulated (*GZMA* and *GZMB*) or were downregulated (GZMH, GNLY and PRF1), compared to HS (P < .05) (Fig 6A). These data suggest expression of cytolytic components may be a unique aspect of the peripheral blood signature of CL, compared to TB and malaria. To explore whether this difference could be explained by cell composition in the peripheral blood mononuclear cell compartment, we used MCP-counter to estimate cell proportions in all three infections. This analysis revealed that the blood of patients with active TB did not show an increased abundance of CTLs and NK cells relative to HS (P>.05), while samples from malaria patients showed significant lower abundances of CTL and NK cells when compared to HS (P < .0001) (Fig 6B). These data suggest that the cytolytic profile we observed is driven by an increased relative abundance of cytotoxic cells in the blood of CL patients, rather than simply increased per-cell expression of these genes.

## Discussion

Given its relative ease of accessibility, peripheral blood offers an appealing sample type for transcriptional profiling to delineate mechanisms of disease. However, the extent to which blood reflects what is happening at a local site is unclear since there have been few direct comparisons of gene expression in tissues and blood collected from the same individuals. Cutaneous leishmaniasis is primarily a localized infection, with little evidence of disease beyond the skin. Nevertheless, we identified two transcriptional signatures in the blood, interferon and cytotoxicity, that were also present in lesions from *L. braziliensis* patients. Both of these

responses are central to the outcome of infection. IFN-γ activates phagocytic cells, such as macrophages and monocytes, to limit parasite replicatioon, while cytotoxicity promotes increased inflammation and thus can exacerbate the disease [6,22,23]. The presence of these signatures in the blood indicates that even in a relatively localized infection systemic changes in gene expression are evident.

A large number of genes are stimulated following exposure to IFNs [44,45]. Precisely which ISGs are expressed is determined by several factors, including the type of IFN, the magnitude of the stimulation, and the cell type responding–all of which are influenced by pathogen species and strain [42,46]. Studies of patients with symptomatic visceral leishmaniasis infected *L. donovani* or *L. infantum* identified IFNG as upregulated in the peripheral blood when compared with healthy controls [25,47,48]. Blood transcriptional ISG signatures have also been described in tuberculosis, leprosy, influenza, Respiratory syncytial virus and malaria [11,14,17,49,50]. In a recent study, the transcriptional profiles from the blood and lungs of C57BL/6 mice were compared after infection with a variety of pathogens known to induce a full spectrum of T cell responses [42], and found that the magnitude of IFN-related signatures was particularly high in the blood of mice infected with *Toxoplasma gondii* when compared to infection by other microorganisms, and this IFN signature was predominately type II in the lungs [42].

In *L. braziliensis* patients the ISG response was biased towards type II ISGs, consistent with the critical role IFN-γ plays in leishmaniasis. Furthermore, we showed that serum levels of IFN-γ correlated with peripheral the peripheral ISG signature. The increased expression of ISGs was driven by an increase in monocytes and macrophages in the blood, suggesting that these cells may already be primed to control the infection prior to lesion entry.

In support of the idea that monocytes entering lesions have been shaped by factors in the blood, CL patients from Corte de Pedra, Brazil show increased frequencies of intermediate and non-classical monocytes in the peripheral blood, combined with increased expression of MHC class II, TNF, and matrix metalloproteinase 9 (MMP-9) [51,52]. Furthermore, cells from patients were better able to kill parasites when compared with cells from healthy subjects [53]. The notion of pre-activation of cells before entry into infected tissues extends to other diseases. For example, a similar response has been described in toxoplasma, where IFN-γ from NK cells in the bone marrow primed monocytes even before entering the blood circulation [54]. A common feature of *L. braziliensis* infections is lymphadenopathy in the lymph nodes draining the site of infection [55]. Parasites and activated T cells are present within these organs, in which the latter are possible sources of the circulating IFN-γ seen in patients [56].

We predict that as a leishmanial infection progresses, this systemic priming of cells will lead to better control of the parasites, promoting more rapid lesion resolution. Given that once activated, macrophages are non-specific, it is possible that other infections dependent on IFN-γ might also be better controlled in these patients. On the other hand, the presence of low levels of IFN-γ leading to a systemic proinflammatory environment could have deleterious effects on patient well-being, a potential negative influence that has yet to be evaluated [57,58]. In contrast to cutaneous leishmaniasis, in visceral leishmaniasis where there is often uncontrolled parasite replication, peripheral blood monocytes were found to have an anti-inflammatory response [59].

Genes specifically induced by type I IFNs were not expressed at higher levels in the blood of patients, suggesting that IFNα/b may not play a systemic role in *L. braziliensis* infections. In lesion biopsies, some type 1-exclusive ISGs were upregulated, although their role in the disease is unclear. Some studies indicate IFN-α/β can play a protective role during the first few days of infection [60], while other studies suggest that type I IFNs suppress the immune responses in both cutaneous and visceral leishmaniasis [61–65]. In active TB, type I IFN signaling has an

important role in driving local susceptibility to the *M. tuberculosis* [66]. Moreover, the blood of active TB patients showed an enrichment for type I-ISG-like signatures, but a negative enrichment for a T cell *IFNG-TBX21* signature [42]. These results together indicate that the class of the infectious pathogen is the main feature that determines the type of systemic IFN-related inflammation [42].

The ability of CTLs and NK cells to kill infected cells is essential for the control of several infectious diseases, although in some cases cytotoxicity is not protective and provokes a destructive inflammatory response [67,68]. This is the case in cutaneous leishmaniasis, where excessive cytolysis leads to inflammasome activation and subsequent production of the proinflammatory cytokine IL-1β [22]. The consequence of this cytolytic response has been shown both in experimental murine models [22,24], as well as in patients where high levels of *PRF1*, *GZMB* and *GNLY* predict treatment failure [19]. Here, we show that elevated expression of these cytolytic genes in patients was not limited to the lesions, since granzymes (*GZMA*, *GZMB*, and *GZMH*), perforin (*PRF1*) and granulysin (*GNLY*) were expressed at higher levels in the peripheral blood of CL patients compared with healthy subjects. Interestingly, VL patients also exhibit higher expression of *GZMA*, *GZMB*, and *PRF1* in the blood compared to healthy subjects, indicating a potential common cytolytic response in leishmaniasis [25]. Correlating with the presence of these cytolytic genes, our analysis of cell abundance indicated that CTLs and NK cells were present at a higher level in the blood of patients compared with healthy subjects. Consistent with these findings, we previously reported that the percentage of NK cells is elevated in patients, with high levels of perforin and granzyme B [69].

The cytotoxicity signature seen in CL was not observed in the blood of either the TB or malaria patients. The cytotoxic function of CD8+ T cells and NK cells has been associated resistance in active TB [70,71]. These cells are found in increased frequencies in the lungs, where they control the proliferation of the bacteria efficiently through direct lyse of infected macrophages upon cytotoxic degranulation [72]. It is possible that the lack of the cytotoxic CTL/NK cell signature in the blood of TB patients is due to the migration of these cells to the lungs. In regards to the malaria infection, as in CL, CTLs play a pathogenic role in the attempt to control the protozoan spread especially in severe cases of the disease such as cases of child infection or cerebral malaria [17,67,68]. Since we only included in this study patients with uncomplicated malaria, this could be the explanation for the lack of a remarkable cytotoxic signature.

In the Northeast of Brazil, the failure rate for the first line drug, pentavalent antimony, can be as high as 50% [73,74], and patients are only given alternative drugs following one or more full treatment courses. Since we found that high levels of cytolytic gene expression in lesions predicts treatment failure [19], one of our goals in this study was to determine if analysis of gene expression in the blood might also be used as a predictor of treatment failure. The elevated levels of cytolytic genes in the blood of patients compared with healthy subjects was evidence of the importance of this pathway in cutaneous leishmaniasis. However, the large variation in expression levels that we observed in lesions from patients was not observed in the blood, which may account for our inability to use cytolytic gene expression in the blood as biomarkers to predict treatment failure.

In the present study, we profiled the transcriptional signatures in the peripheral blood of CL patients and observed an enrichment of an ISG signature associated with monocytes as well as a cytotoxic signature associated with CTLs/NK cells. The immune mechanisms associated with these signatures have a crucial impact on the outcome of infection with *Leishmania*, one leading to better protection while the other promoting increased disease. The consequences for the patients exhibiting these changes in gene expression in the blood has yet to be determined. However, it is reasonable to propose that such responses could influence both

protective and pathologic responses to cutaneous leishmaniasis. Furthermore, these results raised the question of how these systemic responses might influence the response to other infections, as well as the overall health of these individuals.

## Supporting information

**S1 Fig. The interferon signaling is enriched in both lesion and blood from CL patients.** GSEA enrichment plots showing three Interferon-related pathways from the Reactome Pathway Database enriched in the CL peripheral blood (left) and lesion biopsy (right) RNA-seq datasets (IFN signaling, systematic name (sn): M983; IFN gamma signaling, sn: M965; IFN alpha and beta signaling, sn: M973). NES, normalized enrichment score; FDR, false discovery rate.
(TIFF)

**S2 Fig. The expression of cytolytic granule genes in the peripheral blood of CL patients is not correlated with the expression of these genes in the lesions in the same patient.** Correlation between the expression of *GZMB*, *GNLY*, and *PRF1* in peripheral blood and lesions of the same patient. ρ, Spearman's rho correlation coefficient; ns, non-significant P>.05. CPM, counts per million.
(TIFF)

**S3 Fig. MCP-counter estimates cell type abundance in the CL blood bulk RNAseq dataset.** Median of MCP-counter scores for 11 cell types between CL patients (yellow) and HS (blue). Wilcoxon rank-sum test was used for statistical analysis, and P values are represented in the plots.
(TIFF)

**S4 Fig. The expression of genes associated with cytolytic granules correlates with the abundance of CTL and NK cells.** Correlation matrix shows individual correlations between *GZMB*, *GZMA*, *GZMH*, *GNLY*, and *PRF1* genes and the abundance of CTL and NK cells obtained with MCP-counter. The Spearman correlation coefficient ρ is for each correlation is represented in a color scale between -1 and 1 (yellow and red). All correlations included in this matrix were statistically significant, P < .001.
(TIFF)

**S5 Fig. The transcriptional cytotoxicity signature is not predictive of clinical outcome in the peripheral blood of CL patients.** (A) Coefficient of variation (cV) for *GZMB*, *GNLY*, and *PRF1* expression amongst CL patients in the peripheral blood and lesion biopsy datasets (blue and yellow, respectively). (B) Expression of cytolytic genes (*GZMB*, *GNLY*, and *PRF1*) in the peripheral blood of patients who Cured and Failed the first round of treatment. CPM, counts per million log2 scale. (C) Median of MCP-counter abundance scores of T cells, T CD8+ cells, NK cells, and CTL between CL patients who failed the first round of treatment with pentavalent antimony (n = 16, yellow) and CL patients who cured (n = 31, blue). Wilcoxon rank-sum test was used for statistical analysis, and P values are represented in the plots. (D) Principal component analysis showing principal component 1 (PC1) and PC2 for RNA-seq data from the peripheral blood of CL patients who failed or cured the lesion after the first round of treatment with pentavalent antimony. (B) Enrichment score (ES) of the CL peripheral blood signature (51 genes) by ssGSEA in the blood of CL patient who Failed or Cured. ns, non-significant by Wilcoxon rank-sum test, ns, non-significant, P>.05.
(TIFF)

**S6 Fig. *GZMB* and *GZMH* from the CL peripheral blood signature are not enriched in the blood of TB and malaria patients.** A running enrichment score (ES) plot from a GSEA using the 51 genes in the CL peripheral blood signature as a signature in the TB (left) and uncomplicated malaria (right) datasets. The y-axis shows the running ES from each gene from the peripheral blood leishmanial signature, and the x-axis ranks all the genes in the dataset based on their overrepresentation in the different phenotypes of subjects. The total number of genes in each dataset is indicated in parenthesis. Leading-edge genes included in the left side of the blue vertical line (rank at maximum) are overrepresented in the peripheral blood of patients with active TB and malaria. *GZMB* and *GZMH* were highlighted in red the enrichment plots. (TIFF)

## Acknowledgments

We thank Ednaldo Lago, Dr. Luiz Guimarães, and Dr. Edgar M. Carvalho's clinical team in Corte de Pedra, Bahia, Brazil.

## Author Contributions

**Conceptualization:** Camila Farias Amorim, Fernanda O. Novais, Daniel P. Beiting, Phillip Scott.

**Data curation:** Camila Farias Amorim, Ba T. Nguyen, Mauricio T. Nascimento, Jamile Lago, Alexsandro S. Lago.

**Formal analysis:** Camila Farias Amorim, Daniel P. Beiting.

**Funding acquisition:** Lucas P. Carvalho, Phillip Scott.

**Investigation:** Camila Farias Amorim, Fernanda O. Novais, Lucas P. Carvalho, Daniel P. Beiting, Phillip Scott.

**Methodology:** Camila Farias Amorim, Mauricio T. Nascimento, Daniel P. Beiting.

**Project administration:** Phillip Scott.

**Resources:** Phillip Scott.

**Software:** Camila Farias Amorim.

**Supervision:** Fernanda O. Novais, Daniel P. Beiting, Phillip Scott.

**Validation:** Camila Farias Amorim.

**Visualization:** Camila Farias Amorim, Daniel P. Beiting, Phillip Scott.

**Writing – original draft:** Camila Farias Amorim.

**Writing – review & editing:** Camila Farias Amorim, Fernanda O. Novais, Lucas P. Carvalho, Daniel P. Beiting, Phillip Scott.

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
