## [Decision Letter · Decision Letter 0]

21 Jan 2021

Dear Prof. Scott,

Thank you very much for submitting your manuscript "Localized skin inflammation during cutaneous leishmaniasis drives a chronic, systemic IFN-γ signature" for consideration at PLOS Neglected Tropical Diseases. As with all papers reviewed by the journal, your manuscript was reviewed by members of the editorial board and by several independent reviewers. The reviewers appreciated the attention to an important topic. Based on the reviews, we are likely to accept this manuscript for publication, providing that you modify the manuscript according to the review recommendations. 

This is a nice manuscript with solid conclusions. It is important to address the suggestions and comments from both reviewers.

Sincerely,

Dario S. Zamboni, Ph.D.

Associate Editor

Hans-Peter Fuehrer

Deputy Editor

This is a nice manuscript with solid conclusions. It is important to address the suggestions and comments from both reviewers.

Reviewer's Responses to Questions

**Key Review Criteria Required for Acceptance?**

**Methods**

-Are the objectives of the study clearly articulated with a clear testable hypothesis stated?

-Is the study design appropriate to address the stated objectives?

-Is the population clearly described and appropriate for the hypothesis being tested?

-Is the sample size sufficient to ensure adequate power to address the hypothesis being tested?

-Were correct statistical analysis used to support conclusions?

-Are there concerns about ethical or regulatory requirements being met?

Reviewer #1: The study does not representative a clear advancement in the field, since the major findings were already reported elsewhere for a different outcome of the disease (Visceral Leishmaniasis). However, the objectives are clear and the study is well written, and adds to the literature of Leishmaniasis as a whole.

Reviewer #2: The manuscript describes a transcriptomic study of whole blood from patients with L. braziliensis CL and healthy controls. A sub study of lesion biopsies was also performed. It is not clear where healthy control biopsies were obtained from in order to define DEGs? Study design is otherwise clear and appropriate to answer the questions posed. Data analysis is thorough and with no obvious errors. No formal sample size calculations were provided but the statistical analysis was appropriate and sufficient for the conclusions drawn.

**Results**

-Does the analysis presented match the analysis plan?

-Are the results clearly and completely presented?

-Are the figures (Tables, Images) of sufficient quality for clarity?

Reviewer #1: Yes

Reviewer #2: All data are presented in the Figures and supplementary figures are necessary for the story and are of good quality. Interpretation is appropriate and well reasoned.

**Conclusions**

-Are the conclusions supported by the data presented?

-Are the limitations of analysis clearly described?

-Do the authors discuss how these data can be helpful to advance our understanding of the topic under study?

-Is public health relevance addressed?

Reviewer #1: Yes

Reviewer #2: The manuscript draws the straightforward conclusion that that IFN and cytolytic signatures can be identified int he peripheral blood of CL patients. The IFN signature is perhaps not surprising, given data from other diseases, but it is nice to see. The latter appears more restricted to CL at least in comparison to malaria and TB. Data from a viral infection might have made a nice additional comparator. Public health relevance in terms of identifying biomarkers of disease progression in blood and therapeutic intervention strategies are appropriately discussed.

**Editorial and Data Presentation Modifications?**

Reviewer #1: (No Response)

Reviewer #2: The authors should address the following minor points:

1. line 96: for completeness the authors should cite PMID: 28959260 where PBMC from healthy and CL patients was compared using microarray

2. line 120. The source of healthy control biopsies used to provide data for the lesion DEG analysis (line 234) should be mentioned. In addition, please confirm if parasites from these patients were assessed for LRVs?

3. line 133: the authors should mention how and why these 17 were selected and whether this was random or associated with disease status.

4. line 221: Type I and Type II interferons use different receptors -please correct

5. line 435: blood ISG signatures in blood of VL patients should be cited here (PMID: 31419223)

6. Discussion: The authors do not mention that Lbb can be found in lymph node aspirates, concurrently with and indeed before lesion development. Hence, it is not only a skin dwelling parasite and LN residence and local immune responses in the LN may contribute to the systemic signature. I think this is worthy of comment and does not detract from the main message here.

**Summary and General Comments**

Reviewer #1: (No Response)

Reviewer #2: A simple, well conducted study with clear messages of interest to the leishmaniasis research community

PLOS authors have the option to publish the peer review history of their article (what does this mean?). If published, this will include your full peer review and any attached files.

Reviewer #1: No

Reviewer #2: Yes: Paul Kaye
---

## [Decision Letter · Decision Letter 1]

23 Mar 2021

Dear Prof. Scott,

We are pleased to inform you that your manuscript 'Localized skin inflammation during cutaneous leishmaniasis drives a chronic, systemic IFN-γ signature' has been provisionally accepted for publication in PLOS Neglected Tropical Diseases.

Best regards,

Dario S. Zamboni, Ph.D.

Associate Editor

Hans-Peter Fuehrer

Deputy Editor

This is a nice and solid manuscript reporting genetic signatures in localized skin lesions found in leishmaniasis patients. The information will be important to the development of the field. 

Reviewer's Responses to Questions

**Key Review Criteria Required for Acceptance?**

**Methods**

-Are the objectives of the study clearly articulated with a clear testable hypothesis stated?

-Is the study design appropriate to address the stated objectives?

-Is the population clearly described and appropriate for the hypothesis being tested?

-Is the sample size sufficient to ensure adequate power to address the hypothesis being tested?

-Were correct statistical analysis used to support conclusions?

-Are there concerns about ethical or regulatory requirements being met?

Reviewer #1: (No Response)

Reviewer #2: (No Response)

**Results**

-Does the analysis presented match the analysis plan?

-Are the results clearly and completely presented?

-Are the figures (Tables, Images) of sufficient quality for clarity?

Reviewer #1: (No Response)

Reviewer #2: (No Response)

**Conclusions**

-Are the conclusions supported by the data presented?

-Are the limitations of analysis clearly described?

-Do the authors discuss how these data can be helpful to advance our understanding of the topic under study?

-Is public health relevance addressed?

Reviewer #1: (No Response)

Reviewer #2: (No Response)

**Editorial and Data Presentation Modifications?**

Reviewer #1: (No Response)

Reviewer #2: (No Response)

**Summary and General Comments**

Reviewer #1: (No Response)

Reviewer #2: (No Response)

PLOS authors have the option to publish the peer review history of their article (what does this mean?). If published, this will include your full peer review and any attached files.

Reviewer #1: No

Reviewer #2: **Yes: **Paul Kaye

---

## [Editor Report · Acceptance letter]

29 Mar 2021

Dear Prof. Scott,

We are delighted to inform you that your manuscript, "Localized skin inflammation during cutaneous leishmaniasis drives a chronic, systemic IFN-γ signature," has been formally accepted for publication in PLOS Neglected Tropical Diseases.

Best regards,

Shaden Kamhawi

co-Editor-in-Chief

Paul Brindley

co-Editor-in-Chief
